# Towards the Sustainable Exploitation of Salt-Tolerant Plants: Nutritional Characterisation, Phenolics Composition, and Potential Contaminants Analysis of *Salicornia ramosissima* and *Sarcocornia perennis alpini*

**DOI:** 10.3390/molecules28062726

**Published:** 2023-03-17

**Authors:** Maria Lopes, Ana Sanches Silva, Raquel Séndon, Letricia Barbosa-Pereira, Carlos Cavaleiro, Fernando Ramos

**Affiliations:** 1University of Coimbra, Faculty of Pharmacy, Azinhaga de Santa Comba, 3000-548 Coimbra, Portugal or anateress@gmail.com (A.S.S.);; 2Associated Laboratory for Green Chemistry (LAQV) of the Network of Chemistry and Technology (REQUIMTE), R. D. Manuel II, Apartado 55142, 4051-401 Porto, Portugal; 3National Institute for Agricultural and Veterinary Research (INIAV), I.P., Rua dos Lagidos, Lugar da Madalena, 4485-655 Vila do Conde, Portugal; 4Centre for Study in Animal Science (CECA), ICETA, University of Porto, 4051-501 Porto, Portugal; 5Analytical Chemistry, Nutrition and Food Science Department, Pharmacy Faculty, University of Santiago de Compostela, 15782 Santiago de Compostela, Spain; 6Chemical Process Engineering and Forest Products Research Centre, University of Coimbra, Rua Sílvio Lima, Polo II, 3030-790 Coimbra, Portugal

**Keywords:** halophytes, *Salicornia ramosissima*, *Sarcocornia perennis alpini*, nutrients, minerals, phenolic compounds, antioxidants, mycotoxins, heavy metals, food security, food safety

## Abstract

Increasing soil salinisation represents a serious threat to food security, and therefore the exploitation of high-yielding halophytes, such as *Salicornia* and *Sarcocornia*, needs to be considered not merely in arid regions but worldwide. In this study, *Salicornia ramosissima* and *Sarcocornia perennis* alpini were evaluated for nutrients, bioactive compounds, antioxidant capacity, and contaminants. Both were shown to be nutritionally relevant, exhibiting notable levels of crude fibre and ash, i.e., 11.26–15.34 and 39.46–40.41% dry weight (dw), respectively, and the major minerals were Na, K, and Mg. Total phenolics thereof were 67.05 and 38.20 mg of gallic acid equivalents/g extract dw, respectively, mainly *p*-coumaric acid and quercetin. Both species displayed antioxidant capacity, but *S. ramossima* was prominent in both the DPPH and ß-carotene bleaching assays. Aflatoxin B1 was detected in *S. ramosissima*, at 5.21 µg/Kg dw, which may pose a health threat. The Cd and Pb levels in both were low, but the 0.01 mg/Kg Hg in *S. perennis alpini* met the maximum legal limit established for marine species including algae. Both species exhibit high potential for use in the agro-food, cosmetics, and pharmaceutical sectors, but specific regulations and careful cultivation strategies need to be implemented, in order to minimise contamination risks by mycotoxins and heavy metals.

## 1. Introduction

Worldwide, there are about 400,000 plant species [1], of which more than 50,000 are regarded as edible [2]. Yet, of these only about 150 are grown on a large scale for human consumption, 10% of which provide 90% of global dietary energy needs [2]. Clearly, the limited number of crops on which agriculture currently depends contrasts markedly with the entirety of plant bioavailability that the planet offers. This deserves our utmost attention, considering that, in 2020, 10% of the world’s population suffered from severe hunger [3]. In recent years, economic and health crises, conflicts, and the harmful effects of climate change have undermined food security all over the world [3]. Biodiversity can undoubtedly help address challenges such as these, but greater knowledge and a more comprehensive view of available resources are required.

Among the various plant resources currently underutilised, halophytes stand out as relatively rare, i.e., about 1–2% of the total angiosperms, but capable of tolerating salinity levels beyond those with which most plants could cope [4,5]. Climate change is leading to rising sea levels and increased drought stress, which in turn contribute to an increasingly salinised environment. This has tremendously negative consequences for agriculture, as traditional crops cannot withstand severe salinisation and so either brutally reduce their yields or simply wither [4]. By contrast, halophytes are capable of thriving in such adverse conditions, still providing high yields [6]. Moreover, they exhibit a wide range of applications for dealing with severe famine situations, as traditional and gourmet food, animal feed, and as functional ingredients, and finally as sources f bioactive compounds such as polyphenols—compounds of high value due to their antioxidant properties [5]. Despite this, the full potential of halophytes largely remains to be studied and exploited, principally in relation to their role in human nutrition and health. Among all halophytes, *Salicornia* L. and *Sarcocornia* L., both belonging to the Amaranthaceae family, have been singled out as yielding particularly promising food crops, especially owing to their extreme inherent salt tolerance, superior to 500 mM [6]. The two genera comprise about 30 species each, although there is currently no absolute consensus on the exact number of accepted species [7,8]. Actually, *Salicornia* and *Sarcocornia* species are morphologically and ecologically very similar, which makes their distinction complex [9]. The major difference is related to their habit, as *Salicornia* species have an annual life cycle and are of the herbaceous type, whilst *Sarcocornia* species have a perennial life cycle and are woody at least at the base [10]. 

Both exhibit a peculiar appearance, i.e., fleshy, juicy stems with barely perceptible scale-like leaves [9]. An important morphological difference between *Sarcocornia* and *Salicornia* is that the flowers of the former are always arranged in a horizontal row, whilst those of the latter form a triangle with a larger central flower and two smaller lateral ones [9]. *Salicornia* and *Sarcocornia* spp. thrive all over the world, essentially in zones with brackish water, along coastal areas such as salt marshes and mangroves, and in some semi-arid and arid regions such as saline deserts [5,6]. To survive and reproduce in these challenging environments, *Salicornia* and *Sarcocornia*, two extreme dicotyledons, have developed a set of adaptive mechanisms to maintain water and ionic homeostasis and protect against abiotic and biotic stress, including (i) selective ion uptake and transport; (ii) ion compartmentalisation; (iii) succulence; (iv) synthesis and accumulation of osmolytes; and (v) synthesis and accumulation of protective compounds such as phenolic compounds [10]. 

Our study focuses on *Salicornia ramosissima* and *Sarcocornia perennis* subspecies *alpini*, two species widely distributed in European salt marshes, including those along the Portuguese coast. Both have a long history of consumption by coastal communities for seasoning or as substitutes for more conventional vegetables; more recently they have been used as gourmet ingredients in sophisticated salads and dishes. As *Salicornia* and *Sarcocornia* are hard to tell apart, their mixed consumption is frequent, since consumers favour *Salicornia* on account of its more refined flavour and less fibrous texture, whereas, due to its higher yield and year-round availability, producers are partial to *Sarcocornia*. It is important, however, to assess the differences between these two species, namely in their nutritional and phytochemical composition, as well as in terms of consumption safety, to better understand how these plants can contribute to the mitigation of the problems that the world faces today.

Therefore, the present study aims to evaluate the nutritional composition, including the mineral constituents, phenolic profile, and antioxidant potential of *Salicornia ramosissima* and *Sarcocornia perennis alpini* from the Mondego estuary (in the central region of Portugal). The levels of contamination by mycotoxins―aflatoxins (AFs) B1, B2, G1, G2, and ochratoxin A (OTA)—and essential and non-essential heavy metals (HMs)—copper (Cu), zinc (Zn), manganese (Mn), cadmium (Cd), lead (Pb), chromium (Cr), nickel (Ni), cobalt (Co), and mercury (Hg)—were also determined. With these contributions, we aim to increase the knowledge about *Salicornia* and *Sarcocornia*, particularly concerning their potential as a sustainable solution to mitigate food shortages and improve nutrition and health at a global level, while drawing attention to the importance of not neglecting safety aspects. 

## 2. Results and Discussion

### 2.1. Nutritional Composition Analysis

#### 2.1.1. Proximate Composition

The aerial parts of the commonly available but largely underutilised *S. ramosissima* and *S. perennis alpini* were analysed for their nutrient composition. The results are presented in Table 1. 

Moisture content has a major impact on the quality attributes of vegetables. The shoots of both halophytes showed a high moisture content, but *S. ramosissima* excelled in this parameter (89.7 vs. 87.8% fw). For *Salicornia* and *Sarcocornia*, high water content is essential to dilute the high concentration of NaCl present in their tissues and thereby lessen the osmotic pressure, thus allowing them to survive even in extreme saline stress conditions [11]. In a study conducted by Castañeda-Loaiza et al. [11], wild and cultivated specimens of *Sarcocornia fruticosa* were nutritionally characterised, and the cultivated variety exhibited a higher moisture content (see Appendix A), arising from greater water availability. High juiciness is a characteristic that attracts consumers, as it reflects freshness and is associated with a crunchy texture, but it also increases the plants’ susceptibility to both biochemical and microbiological deterioration, for which reason this parameter should be carefully monitored.

The 6.61 and 4.28% dw crude protein values exhibited by *S. ramosissima* and *S. perennis alpini*, respectively, are relatively close to those previously reported for the same species collected in southern Portugal [12] (see Appendix A). Furthermore, in a previous study of ours on the nutritional composition of wild *S. ramosissima* from a salt marsh in the southern arm of the Mondego estuary [8], the identified protein content was lower (see Appendix A), which suggests that factors such as geographic location, as well as soil and water composition, may greatly affect the chemical composition of the plant, as reported by other authors (e.g., [13]). The wild *Salicornia herbacea* from southwestern Tunisia exhibits a remarkable protein content, as reported by Essaidi et al. [14] (see Appendix A). Although the protein levels that we found in *S. ramosissima* and *S. perennis alpini* were modest, these halophytes may still be useful to supplement other sources of protein in human and animal diets. Furthermore, higher protein values can be achieved under optimal growth conditions. By way of example, in the work by Castañeda-Loaiza et al. [11], *S. fruticosa* cultivated in a soilless system under optimised conditions exhibited a protein content higher than that obtained for spontaneous growth specimens (see Appendix A). Likewise, Bertin et al. [13] reported a lower protein content for *Sarcocornia ambigua* from southern Brazil when harvested in its natural habitat, than what was found for the same halophyte when obtained from an experimental crop irrigated with seawater and fertilised with sludge from shrimp production ponds (see Appendix A). Interestingly, Benjamin et al. [15] conducted a study in which the protein profile of the halophytes *Salicornia brachiata* and *Suaeda maritima* was assessed under different concentrations of NaCl (0, 200, 500 mM), and a coordinated, even if distinct, response to salinity was observed in both halophytes. For example, the exposure of *S. brachiata* to 200 mM NaCl induced an up-regulation of several proteins linked to photosynthesis, whereas in *S. maritima* a down-regulation of those proteins was found after addition of 200 and 500 mM NaCl. This is one of the phenomena that may explain the greater salinity tolerance of *S. brachiata* compared with *S. maritima*. Also, proteins related to osmotic and oxidative stress, as well as signalling, transcription, and translation processes were shown to be affected by salinity [15]. Altogether, these data reveal that through the optimisation of growth conditions it should be possible to manipulate, at least partially, the nutritional profile, namely the protein content of *Salicornia* and *Sarcocornia* spp., in accordance with the intended application, so as to maximise its practical potential.

The total lipid content revealed, 1.32% dw in *S. ramosissima* vs. 1.52% dw in *S. perennis alpini*, was within the range of total lipid content reported for most species of *Salicornia* and *Sarcocornia* of spontaneous growth (see Appendix A). In a previous work of ours, the 0.5% dw total lipid levels found for *S. ramosissima* from the southern arm of the Mondego estuary were considerably lower [8]. Although the values obtained in the present study were moderately low for both halophytes, they remain significant when compared with those reported for several commonly consumed vegetables, which have less than 1.0% dw lipid content [16,17]. In the study by Castañeda-Loaiza et al. [11], cultivated *S. fruticosa* specimens exhibited considerably higher lipid content than wild ones (see Appendix A). The same trend was observed in the study by Bertin et al. [13] on *S. ambigua* (see Appendix A). Membrane proteins, whose functionality is strongly dependent on the lipid microenvironment, are a key factor in plants’ salinity tolerance, therefore changes in the lipid profile in accordance with salt concentrations are plausible [18]. In the study by Tsydendambaev et al. [18], the lipid composition of the halophyte *Suaeda altissima* was evaluated under different salinity levels, and the total membrane lipid content in aerial tissues was twice as high in specimens grown under optimal concentrations of 250 mM NaCl than in those grown at concentrations of 1 or 750 mM NaCl. In all, if plants—whether glycophytes or halophytes—are grown under suboptimal conditions, major changes can occur in the lipid composition of their organs. 

The calculated total carbohydrate content—51.3% dw in *S. ramosissima* and 52.3% dw in *S. perennis alpini*—was consistent with reports in other studies (see Appendix A). In plants, soluble carbohydrates produced through the Calvin cycle function as energy sources and components for the synthesis of other organic metabolites. Additionally, they play a major role as signalling molecules involved in immune response, and as osmoprotectants and antioxidants [19,20]. Environmental stress, specifically salt stress, has a harmful effect on the carbohydrate metabolism of vegetal species, and in the case of halophytes, the accumulation of sugars plays a crucial role in osmotic balance, carbon storage, and free radical scavenging [21], in line with the high content of carbohydrates found. Regarding crude fibre, both halophytes showed important levels, but the *S. perennis alpini* 15.3% dw outperformed the *S. ramosissima* 11.2% dw. There is a broad scientific consensus that an adequate intake of dietary fibre offers multiple health benefits, contributing to reduced risk of cardiovascular, metabolic, and digestive diseases through effects such as the lowering of blood pressure and serum cholesterol levels, the improvement of glycaemia and insulin sensitivity, and the regulation of bowel function. Halophytes such as *Salicornia* and *Sarcocornia* can thus play an important role in improving health by contributing to an increase of fibre consumption, at present clearly below the desired levels in developed countries [22]. On the other hand, the fibre content may hinder the consumption of the aforementioned halophytes in the fresh form, since—particularly in their natural habitat—they tend to lose juiciness with the ageing process and develop a marked fibrousness, which in excess discourages consumers. However, this problem does not arise in the case of consumption in the powder form. Halophytes in their wild state, being subjected to greater environmental stress, tend to produce more fibre to increase the strength of the stem [13,23,24]. 

The ash content, that is the total mineral content, was remarkably close to 40% dw for each halophyte. The literature data reveal that the ash levels of *Salicornia* and *Sarcocornia* species are much higher than those reported for common vegetables (see Appendix A). As *Salicornia* and *Sarcocornia* are both obligate halophytes of the accumulator type, they adjust osmotically to soil and water salinity through the accumulation and sequestration of ions in vacuoles, mainly Na and Cl, whilst organic solutes are accumulated in the cytoplasm to avoid deleterious effects on plant metabolism [8,25]. This saline stress tolerance mechanism should explain the high ash content revealed. Regarding the differences between wild and cultivated specimens, it has been observed that *Salicornia* and *Sarcocornia* species from cultivation tend to exhibit higher ash content (see Appendix A). 

Overall, *S. ramosissima* and *S. perennis alpini* displayed energy values of interest, about 244 and 240 kcal/100 g dw, respectively, and proximal composition suitable for human consumption.

#### 2.1.2. Mineral Profile

Owing to the high ash content that characterises *S. ramosissima* and *S. perennis alpini*, knowledge of the composition of their mineral fractions is of particular interest. Table 2 depicts the mineral profile of the investigated halophytes.

The results show that both *S. ramosissima* and *S. perennis alpini* accumulate not only Na, but also other minerals of nutritional interest such as Mg, K, and Ca. Specifically, the mineral accumulation pattern observed in *S. ramosissima* was Na >> Mg > K >> P > Ca, whilst in *S. perennis alpini* it was Na >> K > Mg >> P > Ca. For the different species of *Salicornia* and *Sarcocornia*, Na has been reported as the most abundant mineral, followed by Mg or K, and then Ca and P (see Appendix A). One of the factors shown to have a major influence on the mineral profile of accumulator-type halophytes is the salinity level to which they are exposed in their growing environment. By way of illustration, we refer to the work by Ushakova et al. [26], in which the effect of the NaCl concentration on the mineral composition of *S. europaea* was evaluated, and it was found that as the amount of NaCl increased in the substrate, the levels of K, Mg and Ca in the aerial parts of the plant significantly decreased. The same trend was observed in a more recent study on *S. ramosissima*, carried out by Lima et al. [23]. This phenomenon is related to a mechanism of competitive inhibition [8,27]. Other factors identified as having an important influence on the composition of the mineral fraction of these species include the level of irradiation to which they are subjected [26] and the level of nitrogen fertilisation [28]. In this context, through adjustments in growth conditions, it should be possible to obtain an even more advantageous mineral profile.

### 2.2. Phenolic Compounds and Antioxidant Capacity

#### 2.2.1. Extraction Yield, Total Phenolic Content (TPC), and Total Flavonoid Content (TFC)

The extraction yields for the aerial parts of *S. ramosissima* and *S. perennis alpini* were 14.50% and 13.74%, respectively. The results of the TPC and TFC analysis are shown in Table 3.

Both halophytes had a high content of phenolics, but the *S. ramosissima* extract exhibited a noteworthy TPC of 67.1 mg GAE (256.2 mg ECE)/g of extract dw compared with the 38.2 mg GAE (146 mg ECE)/g of extract dw of *S. perennis alpini*. These values are considerably higher than those reported in the literature for the different *Salicornia* and *Sarcocornia* species (see Appendix A). Analysing the studies by Castañeda-Loaiza et al. [11], Bertin et al. [13], and Riquelme et al. [24], it can be observed that spontaneously growing specimens tend to present TPC values markedly higher than cultivated ones. Furthermore, works such as those by Lima et al. [23], Ventura et al. [29], and Ventura et al. [30] have recounted increases in the phenolic content of *Salicornia* and *Sarcocornia* species with their exposure to increasing levels of salinity, up to a maximum level after which they begin to lose the ability to produce these compounds and/or shift their energy resources to other protection mechanisms. In general, exposure to abiotic and biotic stressors tends to lead to higher TPC values [5]. However, it should be borne in mind that a number of additional factors influence the processes of synthesis, accumulation, and degradation of phenolic compounds, such as the composition of the soil and water, the irrigation regime, the level of exposure to ultraviolet radiation, the life stage and part of the plant analysed, as well as the post-harvest conditions [5,31]. Moreover, the analytical strategy adopted, including the power of the selected extraction solvent, may also affect the TPC results [5,32]. 

Essentially, plant extracts are regarded as of interest when they display a TPC greater than 20 mg GAE/g dw [5]. Considering that the results obtained in our study for *S. perennis alpini* almost doubled that reference value and those for *S. ramosissima* more than tripled it, these extracts can be regarded as highly relevant. Furthermore, the TFC values indicated 186 and 99.3 mg ECE/g extract dw for *S. ramosissima* and *S. perennis alpini*, respectively, revealing the high significance of these extracts, especially *S. ramosissima*. By way of example, the values reported in the literature for green tea (*Camellia sinensis* L.), a plant known for its particular richness in flavonoid compounds, were between 139 and 184 mg ECE/g extract dw [33]. As a general rule, extracts with total flavonoid contents above 100 and 125 mg ECE/g are considered very rich and exceptionally rich [5], respectively, and therefore of high application value in the food, pharmaceutical, and cosmetic industries.

#### 2.2.2. Phenolic Profile

Since the TPC estimated by the Folin–Ciocalteu assay and the TFC estimated by the aluminium chloride method do not provide a complete overview of the quantity and quality of the different phenolic constituents, the adoption of more advanced analytical techniques―such as chromatographic ones―is crucial to obtain more information about the individual phenolic components. Table 4 describes the phenolic compounds identified by UHPLC-ESI-MS/MS in the extracts of *S. ramosissima* and *S. perennis alpini*.

The identification was achieved by assessing the elemental composition data determined from mass measurements in negative ionisation mode, in comparison with data available in the literature and those obtained from the available standards. Exceptionally, compound 25 (quercetin) was measured in the positive mode (*m*/*z* 274), as described by Andrade et al. [39]. Each compound was characterised by its retention time (t_r_), maximum absorption wavelengths (λ_max_), structural class, molecular formula, molecular ion, and main MS/MS fragments. A total of 33 compounds were identified in the extracts, belonging to the families of phenolic acids and flavonoids: among the first category were hydroxybenzoic acids (compound 1) and esters of hydroxybenzoic acids with quinic acid (compounds 2 and 11), hydroxycinnamic acids (compounds 4, 10 and 12), hydroxycinnamic acid glycosides (compounds 6 and 8) and esters of hydroxycinnamic acids with quinic acid, also named chlorogenic acids (compounds 3, 5, 7 and 9); in the latter, we refer to flavonols (myricetin, quercetin, kaempferol, and isorhamnetin) and their glycosides (compounds 13, 14, 15, 16, 18, 20, 21 and 22), flavanones (naringenin) and their glycosides (compounds 16, 19 and 33), flavones (luteolin and apigenin) and their glycoside derivates (compounds 18, 23 and 32) and flavanols (B-type proanthocyanidins). Almost all of these compounds were found in both analysed extracts, except for compounds 1, 9, 13, 19, 22, and 30, which were tentatively identified only in the *S. ramosissima* extract, while compounds 11 and 31 were detected in only in the *S. perennis alpini* extract. However, considering the content of the main phenolic compounds identified and quantified, some differences were observed in the extracts analysed. From the 33 chemical compounds described in Table 4, 12 polyphenols were confirmed by the commercial standards and were quantified in both *S. ramosissima* and *S. perennis alpini* extracts according to the parameters described in Table 5.

*S. ramosissima* extract exhibited a higher total amount of phenolic compounds, which is in agreement with the results obtained by the Folin–Ciocalteu method. Among the extracts analysed, *S. ramosissima* presented a higher proportion of phenolic acids than flavonoids, whereas in the *S. perennis alpini* extract flavonoids were the principal polyphenols quantified. The results revealed that *p*-coumaric acid and chlorogenic acid were the most abundant phenolic acids in both extracts, while the most abundant flavonoids were quercetin and rutin. *S. ramosissima* extract also contained high amounts of kaempferol when compared with *S. perennis alpini* extract.

Regarding the literature data, we refer, by way of example, to the study conducted in *S. fruticosa* by Castañeda-Loaiza et al. [11], according to which the main compounds detected were chlorogenic acid, catechin hydrate, and 3,4-dihydroxybenzoic acid, the cultivated specimens displaying in a higher content of phenolic compounds than the wild ones. The content of phenolic compounds determined in that study [11] for both cultivated and wild *S. fruticosa* was generally higher than that obtained in the extract of *S. perennis alpini* in the present study, although the caffeic acid, *p*-coumaric acid, rutin, and quercetin levels were substantially lower. As for *S. ramosissima*, it is worth noting the study by Silva et al. [40], which reported myricetin, gallic, ferulic, and protocatechuic acids, and catechin as major phenolics in a sample of this species from the Ria de Aveiro, central Portugal. It should be noted that in the said study [40], the content of the identified phenolic compounds was markedly lower than that obtained in the present research for *S. ramosissima* from the Mondego estuary. Altogether, it is clear that there is important intra- and inter-species variability within the phenolic profiles of *Salicornia* and *Sarcocornia*. All things considered, our results indicate that *S. ramosissima* and *S. perennis alpini* have not only important nutritional value but also numerous bioactive compounds, namely of the polyphenol type. These molecules are particularly interesting owing to the beneficial antioxidant properties that have been attributed to them, which are highly relevant for the plant species they are part of as well as for those who consume them.

#### 2.2.3. Antioxidant Capacity

Antioxidants have been reported to have a major role in the prevention and mitigation of a multitude of pathologies, through their ability to protect organisms from the excessive production of free radicals [5]. Furthermore, in the food and dermo-pharmaceutical industries, there is currently a growing interest in the replacement of synthetic antioxidants by natural ones, as the latter tend to be regarded as safer [5]. In the present work, the antioxidant potential of the halophytic species under study was evaluated by both DPPH radical scavenging activity assay and the ß-carotene bleaching inhibition test, and the results are summarised in Table 6. Note that the combination of different methods for evaluating antioxidant capacity should provide more reliable results than a single method alone, since oxidative stress is produced by the action of several reactive species, which have different reaction mechanisms [5].

In the obtained results, *S. ramosissima*, which presented the highest content of phenolics and flavonoids, also exhibited the highest antioxidant capacity measured by the DPPH assay, 30.2 mg TE/g extract dw (vs. 11.0 for *S. perennis alpini*), and by the ß-carotene method with an AAC of almost 1700 (vs. 1403 for *S. perennis alpini*).

Our findings are in line with several research studies that have demonstrated the antioxidant capacity of extracts from different *Salicornia* and *Sarcocornia* species, such as those conducted by Barreira et al. [12], Essaidi et al. [14], Clavel-Coibrié et al. [41], and Cho et al. [42]. These authors attributed these activities mainly to the content of phenolic acids and flavonoids, which is consistent with our results. However, it should be noted that the global antioxidant capacity in extracts may depend not only on the quantity of these molecules, but also on the variability of their chemical structures, as well as on their synergistic or antagonistic interactions [5,43]. Additionally, the presence of other antioxidant compounds may also influence the total antioxidant capacity of the extracts [5].

### 2.3. Contaminants

#### 2.3.1. Mycotoxins

Mycotoxins are a series of secondary metabolites produced by a variety of fungi that grow on plant products, either in the field or during transport, processing, and storage [44]. For spices and seasoning vegetables, there are two groups of mycotoxins of major concern: AFs and OTA [44]. AFs are produced by some *Aspergillus* species and are considered the most harmful class of mycotoxins [45]. The most important members of this family, i.e., AFB1, AFB2, AFG1, and AFG2, in addition to being hepatotoxic and immunotoxic, have been classified as group I carcinogens by the International Agency for Research on Cancer (IARC) [45,46]. OTA is produced by *Aspergillus* and *Penicillium* species and is classified as a probable human carcinogen (group 2B) [45]. Thus, owing to their high toxicity and thermostability, several countries have set limits for the occurrence of these compounds in various foods intended for human or animal consumption, although such limits are yet to be established for halophyte plants [44].

In the present research work, the determination of AFs and OTA levels was carried out in the two target halophytes to assess the risk of contamination, and the results are shown in Table 7.

Important levels of contamination by AFB1 were found in *S. ramosissima*, that is, 5.21 μg/Kg dw, whereas no AF contamination was observed in *S. perennis alpini*, and no OTA contamination was detected in any sample of either species. The risk of contamination by mycotoxins in halophytes was reported in a previous study of ours, in which samples of different species of *Salicornia* exhibited levels of total AF contamination greater than 10 µg/Kg [44]. The European Commission, through regulation no. 1881/2006 and its amendment no. 165/2010, stipulates maximum levels of AFs allowed in some spices (*Capsicum* spp., *Piper* spp., *Myristica fragrans*, *Zingiber officinale*, *Curcuma longa*, and their mixtures) as 5 µg/Kg for AFB1 and 10 µg/Kg for the total AFB1, AFB2, AFG1, and AFG2 [47]. Therefore, we may assume that contamination levels higher than these in halophytes should also represent a threat to public health. To guarantee the safe consumption of halophytes, it is on the one hand utterly crucial that the legislation specifically considers this type of food, while on the other hand producers are duly trained to adopt measures that lessen the contamination risk, including elimination of diseased specimens, avoidance of contact of shoots with the soil, rinsing with potable water, transportation under refrigeration, processing under hygienic conditions, and prevention of prolonged storage, as discussed in more detail in Lopes et al. [44]. Furthermore, it will be important to conduct further studies to explain the differences in the levels of contamination in the two halophytes studied, given that both species have the same provenance and were subject to the same processing in identical conditions. One hypothesis is that *S. perennis alpini* may be less susceptible to contamination by mycotoxigenic fungi because it is a more fibrous species and has a stronger cell wall, which creates an improved barrier against invasion by pathogenic microorganisms.

Another aspect that deserves our attention in this context is the relationship between phenolic content and mycotoxin contamination. Phenolics, in particular phenolic acids and flavonoids, have been reported to exhibit important antifungal properties in several plant species and against a wide variety of pathogenic fungi (e.g., [48,49]). Moreover, various studies have shown the ability of these compounds to inhibit the biosynthesis of various mycotoxins, namely AFs (e.g., [50,51,52]). Consistent with this are the findings of the study conducted on rice by Giorni et al. [53], in which it was observed that the TPC and phenolic profile varied significantly depending on the level of fungal infestation and the presence of mycotoxins. In particular, the authors of that study reported that the phenolic content tends to be higher in the early stages of plant development and decrease during the growing season; however, when a fungal infestation occurs, the decrease in phenolic levels is less noticeable, attributed to the fact that the plant needs a high content of these compounds to defend itself against the fungal infection [53]. Bearing this in mind, the phenolic content of the halophytes under analysis may have been affected by the level of exposure to mycotoxigenic fungi, particularly in the case of *S. ramosissima* which was considerably contaminated by AFB1.

#### 2.3.2. Essential and Non-Essential Heavy Metals

The definition of the term “HM” is still controversial [54,55]. According to Csuros and Csuros [56], HM designates a metal with a density above 5 g/cm^3^. Ali and Khan [54] proposed a broader definition, in which HMs are considered naturally occurring metallic elements with atomic numbers greater than 20 and elemental densities above 5 g/cm^3^. This latter definition yields a total of 51 elements to be designated as HMs [54]. By this definition, the term HMs should not necessarily be associated with pollution nor toxicity [54]. Yet, analysing the literature, it is observed that the term HM is often used as a group name for metals and metalloids that have been associated with contamination and (eco)toxicity [57]. Clearly, an unambiguous definition of the term HM is sorely needed, but while such is not available and for the purposes of the present manuscript, the definition of HM based on both atomic number and element density proposed by Ali and Khan [54] is adopted here.

HMs can be divided into two main groups: (1) essential and (2) non-essential. The former includes elements of biological relevance, i.e., Fe, Zn, Cu, Mn, and Co, which function as protein cofactors in a wide range of biological processes. These, not being toxic when present in the trace amounts required by the organism, can induce toxicity beyond a certain limit [58,59]. Essential HMs are also commonly referred to as essential trace elements. Non-essential HMs, such as Cd, Pb, and Hg, have no known biological function and are toxic even at lower levels of exposure [58,59]. The latter are considered a particularly worrying category of contaminants owing to their toxic, non-biodegradable and bioaccumulative character [58,60]. While naturally present in the environment in trace concentrations, pollution caused by anthropogenic activities has greatly intensified HMs’ presence in ecosystems, making them a major threat to global health [58,59].

In plants, excessive concentrations of HMs can cause damage through phenomena such as hyper-generation of reactive oxygen species, disruption of enzymatic processes, alteration of membrane permeability, inactivation of photosystems, and disturbance of mineral metabolism [58,61]. In animals, the consequences of the accumulation of HMs have been well studied, with reported effects such as oxidative stress and inflammation (e.g., Pb, Cd, and Ni) [62,63], changes in gene expression (e.g., Pb, Cd, and Mn) [64,65], destruction of the mucosa of the intestinal tract, and changes in the microbiota (e.g., Pb) [66]. Toxicity occurs through processes such as the inhibition of antioxidant enzymes, substitution of native metal ions in enzymes involved in metabolic processes, disruption of protein structures, inhibition of DNA repair, and formation of protein and/or DNA cross-links [60,67]. Hence, consumption of food and water contaminated by HMs can lead to serious neuronal, hepatic, renal, immunological, cardiovascular, reproductive, and gastric damage, as has been recounted in multiple studies [68].

Coastal wetlands are included among the most polluted ecosystems [69]. They are subject to high input of materials from adjacent environments, including HMs from various terrestrial, marine, and atmospheric sources [69]. Activities such as agriculture, aquaculture, sewage discharge, transportation, and oil spills are considered the main causes of HMs deposition in these areas [69]. Upon reaching the salt marshes, these contaminants spread and interact with the local biota community [70]. Halophytes are highly resistant, not only to salinity but also to HMs, and this resistance depends at least partly on common protective mechanisms such as the synthesis of phenolics [71]. The main route of uptake of HMs for most halophytes is through the root system, with subsequent translocation to the aerial parts. In general, most toxic metals tend to accumulate in the roots and do not reach the shoots, which are the more frequently consumed parts of the plant. However, it should be noted that there is great variability in the uptake rate, depending on both the HMs and halophytes involved [71].

The results of the HMs analysis of *S. ramosissima* and *S. perennis alpini* samples are summarised in Table 8.

Cu, Cd, Pb, and Cr levels did not vary significantly between the two species under study, whilst significant variation was observed for Zn, Mn, Ni, Co, and Hg levels (*p* < 0.05). Barreira et al. [12] also reported an important variation in the concentration of the different HMs in *Salicornia* and *Sarcocornia* species, even when differences in environmental factors were minimal. These differences may be related to factors such as, among others: (i) differences in the life cycle, as *S. ramosissima* is annual and *S. perennis alpini* is perennial, and as a consequence, the metabolic requirements for certain macronutrients and micronutrients at the time of harvest may differ; (ii) the presence of root exudates, which can improve the solubility of certain HMs and thus increase their accumulation [72]; and (iii) differences in the microbiome [73]. The order of HM concentrations determined in this study for *S. ramosissima* was Mn > Zn > Cr > Cu > Co > Ni > Cd > Pb > Hg, whilst that for *S. perennis alpini* was Mn > Zn > Ni > Cr > Cu > Co > Pb > Cd > Hg. *S. ramosissima* showed a slightly higher total concentration of the HMs analysed, mainly due to the higher Mn content. In this regard, *S. ramosissima* was previously identified by Barreira et al. [12] as an excellent source of this trace mineral, which has a major role in amino acid, carbohydrate, and lipid metabolism, and bone development, among other functions [74]. By way of example, the consumption of 100 g of *S. ramosissima* shoots from the Mondego estuary would mean an intake of about 0.7 mg of Mn, and the World Health Organisation recommends a consumption of 2 to 3 mg/day [75]. This capacity for Mn accumulation might pose a health risk if the plant’s growth environment is heavily contaminated with this element, given that it can induce neurotoxicity at excessive levels [74,75]. Moreover, Mn, which until now has gone unnoticed as a contaminant due to its role as a micronutrient and its ubiquity in the environment, has recently gained prominence owing to its presence in ecosystems at increasing levels resulting from human activities [75]. Another important difference between samples was observed in their comparative levels of Ni accumulation, almost twice as high in *S. perennis alpini*. Note that foods contain on average less than 500 μg/Kg of Ni [76], thus the 1750 μg/Kg determined for *S. perennis alpini* should be borne in mind, even if it is within reported levels for more conventional vegetables regarded as Ni accumulators [76]. Although the extent of the toxic effects resulting from the consumption of Ni-contaminated food is not yet well established, caution is advised, as it has consistently been shown to play an important role in inducing oxidative stress [76]. In addition, it is important to point out that exposure to Ni is a major cause of contact dermatitis, and for some individuals the allergic reaction can occur even at very low levels of exposure [77]. Therefore, the adoption of protective measures such as the use of gloves is strongly recommended when handling food contaminated by this HM. Likewise, the higher level of Hg accumulation in *S. perennis alpini* compared with *S. ramosissima* is also worth mentioning, mainly due to the risk of neurological damage caused by exposure to high concentrations of this element [68].

As a whole, the data indicate that the levels of HMs in *S. ramosissima* and *S. perennis alpini* are acceptable. In particular, the content of Cd and Pb are well below the maximum established in the European Union through Regulation no. 1881/2006 and amendments, Regulation no. 2021/1317, and Regulation no. 2021/1323, i.e., 0.2 [78] and 0.3 [79] mg/Kg fw, respectively. The level of Hg detected in *S. perennis alpini* matches the maximum limit allowed for algae according to Regulation no. 396/2005, i.e., 0.01 mg/Kg [80]. We emphasise that this regulation defines the maximum levels of pesticide residues allowed in different categories of foodstuffs, and that, owing to the absence of a more specific regulation for halophytes or even for algae, we use this as the basis for discussion, similarly to other authors [81,82]. In this context, it is furthermore important to note that the Mondego estuary is considered a less polluted estuary, due to the limited industrialisation of the area [83,84]. However, with climate change, phenomena such as forest fires have intensified dramatically in Portugal, and a direct relationship has been documented between the mobilisation and accumulation of elements such as Hg, Mn, and Zn in ecosystems and the occurrence of these events [84,85], which could ultimately lead to a greater accumulation of these HMs in the species under study. Finally, as shown in the studies by Yang et al. [86] and Sanjosé et al. [72], in the case of specimens from highly contaminated areas, the risk of accumulation by HMs at levels that represent a health risk is real and should not be overlooked. Hence, the introduction of specific legislation for halophytes is a fundamental step to guarantee the safety of their consumption.

## 3. Materials and Methods

### 3.1. Sampling Area

The Mondego estuary comprises a 16 km^2^ polyhaline and well-mixed mesotidal system located on the Atlantic coast, that benefits from a temperate climate [87]. It consists of northern (deeper and more hydrodynamic) and southern arms (shallower with extensive intertidal mudflats), separated by an island—Morraceira—the provenance of the samples analysed in this study [88]. This estuarine system supports industrial activities such as fishery, salt production, and aquaculture [87]. In total, it comprises an area of 8452 km^2^ of salt marshes [88], an important part of which is abandoned land, contributing to its degradation and the alteration of ecological conditions. Thus, the cultivation of halophyte species can be an important rehabilitation strategy for these saline areas, and make an important contribution to the local economy.

### 3.2. Sample Collection and Preparation

Specimens of *S. ramosissima* (Figure 1a) and *S. perennis alpini* (Figure 1b) were obtained from a small local producer on the aforementioned Morraceira island (Mondego estuary), who collects the specimens that grow in his salt marsh. The identification of the said species was conducted by a taxonomy specialist based on the criteria of Valdés and Castroviejo [89]. The harvest was carried out manually using gloves, during July 2021. July is the time when the first adult *Salicornia* specimens can be collected, whilst November typically marks the onset of the senescence phase of this species, in other words, the end of its life cycle. *Sarcocornia*, being an annual species, can be harvested at any time of the year. Traditionally, both species tend to be collected in the spring and summer months, when their consumption is highest. Regarding the harvesting process for this research work, only specimens of identical maturity, regular size, and healthy appearance were selected, those with a less than ideal appearance being rejected. Still at the producer location, the plants were washed with marsh water to remove sediment and other foreign materials. The aerial parts were then separated and placed in hygienised plastic boxes, and the remaining plant material discarded. In order to obtain results comparable with those common in the sector, the aforementioned procedure was identical to the method usually adopted by small producers. Transport between the collection and sample analysis sites took 45 min under ordinary conditions. On arrival at the laboratory, the plant material was carefully washed with deionised water, externally dried with a paper towel and dehydrated or immediately frozen, depending on subsequent analysis. The dried samples were dehydrated at 40 °C ± 5 °C for 35 h in an oven, ground to a granulometry ≤ 1 mm, thoroughly homogenised, and stored at 4 °C in sterile, airtight plastic containers while awaiting chemical analysis. The frozen samples were first milled, homogenised, and then stored at −20 °C in the same type of container. Dried samples were utilised to determine centesimal composition (except moisture), mineral profile, and mycotoxin and HM contamination; frozen samples were employed to evaluate the phenolic composition and antioxidant capacity; and, finally, fresh samples were used only for moisture content analysis.

### 3.3. Chemicals and Reagents

Reagents used for proximate composition analysis were Kjeldahl tablets (3.5 g potassium sulphate plus 3.5 mg selenium) from Foss (*Höganäs, Sedwen)*, sulphuric acid (H_2_SO_4_) and sodium hydroxide (NaOH) from PanReac AppliChem ITW Reagents (Darmstadt, Germany), methyl red (C_15_H_15_N_3_O_2_) and saccharose (C_12_H_22_O_11_) from Merck (Darmstadt, Germany), and petroleum ether (C_6_H_14_) from VWR Chemicals (Fontenay-sous-Bois, France). For the evaluation of the mineral profile, standard solutions of sodium (Na) at a concentration of 10,000 mg/L, and of potassium (K), calcium (Ca), phosphorous (P), and magnesium (Mg) at concentrations of 1000 mg/L were purchased from Sigma-Aldrich (Madrid, Spain). The calibration curves were prepared with the aforementioned standard stock solutions diluted with ultrapure deionised water obtained using a Milli-Q water purification system from Millipore (Bedford, MA, USA). The measurement of the target minerals was performed on an Architect ci8200 chemical auto-analyser, with an Abbott reagent kit (Chicago, IL, USA). For the total phenolic content and antioxidant capacity assessment, the reagents used were absolute ethanol, Folin & Cioucalteu’s phenol reagent, gallic acid, 2,2-diphenyl-1-picrylhydrazyl (DPPH) (C_18_H_12_N_5_O_6_), 6-hydroxy-2,5,7,8-tetramethylchroman-2-carboxylic acid (trolox) (97% purity), linoleic acid (C_18_H_32_O_2_), and polyoxyethylene sorbitan monopalmitate (Tween^®^ 40) (C_62_H_122_O_26_) from Sigma-Aldrich (Steinheim, Germany), sodium carbonate (Na_2_CO_3_) and β-carotene from Fluka Chemicals (Sintra, Portugal), and methanol from VWR Chemicals (Fontenay-sous-Bois, France). For determination of the total content of flavonoids, the reagents used were sodium nitrite (NNaO_2_) and sodium hydroxide (NaOH) both from Merck (Darmstadt, Germany), aluminium chloride (AlCl_3_) from Fluka Chemicals (Sintra, Portugal), and epicatechin from Sigma-Aldrich (Madrid, Spain). For the identification and quantification of the phenolic compounds, standards of protocatechuic acid, caffeic acid, ferulic acid, *p*-coumaric acid, rutin, myrecetin, naringenin, luteolin, kaempferol, and apigenin were purchased from Sigma-Aldrich (Madrid, Spain), and the standards of chlorogenic acid and quercetin were provided by Fluka Chemie AG (Buchs, Switzerland). The standard stock solutions of the aforementioned phenolics were each prepared at a concentration of 1000 mg/L using methanol, which was obtained from Merck (Darmstadt, Germany). Regarding the study of the occurrence of mycotoxins, the standards of the AFs, OTA, and zearalanone (ZAN) were supplied by Sigma-Aldrich (Madrid, Spain). The standard stock solutions for AFB1, AFG2, and OTA were prepared via dissolution in methanol, and those for AFB2, AFG1, and ZAN were prepared in acetonitrile. Acetonitrile and methanol were from Merck (Darmstadt, Germany), as well as formic acid (CH_2_O_2_) used in the mobile phase. For the determination of the HMs levels, the hydrochloric acid (HCl) (37%, *v*/*v*) used for ash dissolution was from Chem-Lab NV (Zedelgem, Belgium) and the standard stock solutions for each target element were from Merck (Darmstadt, Germany), with the exception of iron (Fe) which was purchased from Sigma-Aldrich (Madrid, Spain).

### 3.4. Nutritional Composition Analysis

#### 3.4.1. Proximate Composition

The aerial parts of *S. ramosissima* and *S. perennis alpini* were analysed for their nutritional content, following the Association of Official Analytical Chemists (AOAC) procedures [90]. Moisture was determined by measuring the amount of water removed from the fresh samples after direct heating in a forced air circulation oven at 105 °C until a constant weight was achieved (AOAC 934.01). Ash was determined via incineration of the samples in a muffle furnace at 550 °C for 7 h (AOAC 930.05). Total lipids were quantified by the Soxhlet extraction method (AOAC 991.36). Crude fibre was quantified using a F-6P fibre extraction unit (Raypa, Barcelona, Spain) through successive hydrolysis with 100 °C 0.26 N sulphuric acid and 0.32 N sodium hydroxide for 30 min each (AOAC 962.09). Crude protein content was calculated by multiplying the nitrogen (N) value by a conversion factor of 6.25. Nitrogen was quantified via the Kjeldahl procedure (AOAC, 2000.11). Total carbohydrates were obtained by difference. Energy was estimated using the Atwater conversion factors, i.e., carbohydrates: 4.0 kcal/g; proteins: 4.0 kcal/g, and lipids: 9.0 kcal/g.

The ash, total lipids, crude fibre, and crude protein contents were expressed as percentages of dry weight (% dw), whilst moisture content was expressed as a percentage of fresh weight (% fw).

#### 3.4.2. Mineral Profile

The extraction and quantification of minerals from *S. ramosissima* and *S. perennis alpini* was performed according to the “green” method by Lopes et al. [8]. Briefly, 200 mL of deionised water were added to 6.0 ± 0.5 g of each powdered sample, and the mixtures were placed under sonication in a 35 kHz ultrasound bath for an optimised time of 5 min. Thereafter, the solutions were homogenised and filtered through 0.45 µm nylon membranes. Sample processing was performed on an Architect ci8200 chemical autoanalyser (Abbott Laboratories, Chicago, IL, USA); Na and K concentrations were determined by indirect potentiometry, and those of Ca, Mg, and P by photometry. The results obtained are expressed in milligrams per gram of plant (aerial parts) on a dry weight basis (mg/Kg dw).

### 3.5. Phenolic Compounds and Antioxidant Capacity

#### 3.5.1. Extraction Procedure

Previously crushed and frozen samples of *S. ramosissima* and *S. perennis alpini* were thawed at room temperature (≈23 °C). Thereafter, the extraction procedure was performed following the method described by Robalo et al. [91]. Briefly, 5 ± 0.5 g of each sample was transferred to a Falcon tube, and 50 mL of absolute ethanol was added. The tubes containing the mixtures were placed on a shaker at 400 rotations per minute (rpm) at room temperature for 30 min, and then centrifuged at 5000× *g* at 15 °C for 15 min. Afterwards, the resulting supernatant was collected and transferred to a pyriform flask, and the ethanol evaporated at 35 °C until dry. The extracts were scraped from the flasks and placed in sterile, airtight tubes, and stored at 5 °C until further use.

The extracts yield (%) was calculated using Equation (1):(1)$$\mathrm{Yield}=\frac{\mathrm{Em}}{\mathrm{Sm}}\times 100$$

Em represents the mass of the obtained extract after solvent evaporation and Sm the mass of the fresh plant sample used for the extraction process.

Ethanolic solutions with final concentrations of 5 mg/mL of each extract were prepared in order to evaluate their total content of phenolic compounds and antioxidant capacity. For the study of the detailed phenolic profile of these extracts, methanolic solutions were prepared at a final concentration of 5 mg/mL for each.

#### 3.5.2. Total Phenolic Content (TPC)

The determination of the TPC was carried out through the Folin–Ciocalteu colourimetric method described by Singleton et al. [92]. Thus, 7.5 mL of the Folin–Ciocalteu reagent (1:10, *v*/*v*) was added to a 1 mL aliquot of each plant sample extract and the mixtures were then homogenised. After 5 min of incubation, 7.5 mL of an aqueous solution of sodium carbonate (60 mg/mL) was added. Thereafter, the reaction mixtures were vortexed and allowed to stand for colour development in the dark at room temperature for 2 h. After the reaction period, the absorbance was measured at 725 nm. A standard calibration curve was plotted using different concentrations of gallic acid (y = 7.8057x + 0.0109; r^2^ = 0.9986; range: 5–100 µg/mL). The TPC was expressed as milligrams of gallic acid equivalent per gram of extract dry weight (mg GAE/g extract dw).

#### 3.5.3. Total Flavonoid Content (TFC)

The measurement of the TFC was conducted according to the methodology proposed by Yoo et al. [93]. In brief, 1 mL of each plant extract was mixed with 4 mL of ultrapure water and 0.3 mL of an aqueous solution of sodium nitrite (5%, *w*/*v*), followed by an incubation period of 5 min and the addition of 0.6 mL of aluminium chloride (10%, *w*/*v*). After a further 6 min incubation period, 2 mL of sodium hydroxide (1 M, *w*/*v*) and 2.1 mL of ultrapure water were added, and the solution was homogenised. Finally, the absorbance of the reaction mixture was determined at 510 nm. A calibration curve was plotted using different concentrations of epicatechin (y = 2.0304x + 0.017; r^2^ = 0.997; range: 5–200 µg/mL). The TFC results are expressed in milligrams of epicatechin equivalents per gram of extract dry weight (mg ECE/g extract dw).

#### 3.5.4. Phenolic Profile

The identification and quantification of phenolic compounds in the extracts were conducted using an UHPLC-ESI-PDA-MS/MS (Thermo Fisher Scientific, San José, CA, USA), equipped with an Accela quaternary pump, a degasser, an autosampler, a column oven, and a photodiode array detector (PAD), coupled to a triple quadrupole mass spectrometer TSQ Quantum Access MAX and an electrospray ionisation source (ESI). The instrument control and data collection and processing were performed with Xcalibur 2.1 software (Thermo Fisher Scientific, San José, CA, USA). The phenolic compounds were separated on a reverse phase Kinetex EVO C18 100 Å column (150 × 3 mm, 5 µm particle size) (Phenomenex, Torrance, CA, USA), thermostatically set at 30 °C. The injection volume was 10 μL. The mobile phase was composed of two solvents: water with 0.1% formic acid (solvent A) and methanol with 0.1% formic acid (solvent B). The flow of the mobile phase was set at 0.6 mL/min and the gradient elution method applied was as follows: 95% solvent A; 3 min, 90% solvent A; 10 min, 80% solvent A; 18 min, 70% solvent A; 25 min, 30% solvent A; 33 min, 0% solvent A; 33–40 min, 0% solvent A and 100% solvent B isocratic; and finally, the column was washed and reconditioned with 95% solvent A (40–46 min). PDA spectra acquisition was performed continuously using a full scan modality during the run in the range of 200 to 600 nm. The mass spectrometer electrospray ionisation source was operated in both negative and positive modes, according to the nature of the phenolic compounds. The optimised MS/MS conditions were as follows: electrospray voltage: 2.5 kV; vaporiser and capillary temperatures: 340 °C and 350 °C, respectively; sheath gas pressure: 25 psi, and auxiliary gas pressure: 5 arbitrary units. Nitrogen (purity > 99.98%) was used as a sheath gas, ion sweep gas, and auxiliary gas, and argon was the collision gas (1.5 mTorr). After an initial screening in the MS scan range of 100–800 m/z, tentative identification of polyphenols was accomplished by comparing their precursor ions [M]- or [M-H]+ (depending on the phenolic) and mass spectrometry fragmentation patterns (MS/MS) with those previously described in the literature (see Table 4 and Table 5). The MS/MS data acquisition was performed in single reaction monitoring (SRM) mode. The confirmation of individual phenolic compounds’ identity was achieved by comparing the retention times, UV–Vis spectra (λ_max_), and MS/MS data with those obtained by injecting commercial standards under the same HPLC conditions. Quantification was performed by the external standard method with six-point calibration curves, using the most abundant fragments in SRM mode acquisition.

#### 3.5.5. Antioxidant Capacity

##### DPPH (2,2-Diphenyl-1-Picryl-Hydrazyl) Radical Scavenging Assay

The ability of *S. ramosissima* and *S. perennis alpini* extracts to scavenge DPPH free radicals was evaluated according to the procedure described by Martins et al. [33]. Briefly, 2 mL of a freshly prepared methanolic solution of DPPH (14.2 µg/mL) was added to 50 µL of each ethanolic plant extract solution (5 mg/mL), and the mixture was appropriately homogenised and incubated in the dark at room temperature for 30 min. For the control assays, 50 µL of ethanol and 2 mL of the DPPH solution were used. Afterwards, the absorbance of the resulting solution was measured at 515 nm. A calibration curve was drawn using different concentrations of trolox (y = 0.6242x + 0.467; r^2^ = 0.9984; range: 10–150 µg/mL), and the results were expressed as milligrams of trolox equivalent per gram of extract dry weight (mg TE/g extract dw).

##### β-Carotene Bleaching Assay

The procedure was carried out as described by Martins et al. [33]. First, an emulsion was prepared, in which 1 mL of a solution of β-carotene in chloroform (2 mg/mL) was added to 20 mg of linoleic acid and 200 mg of Tween^®^ 40 in a round-bottomed flask. Then, chloroform was removed in a rotary evaporator at 40 °C. Then, 50 mL of ultrapure water saturated with oxygen was added to the obtained residue and the mixture was repeatedly shaken to form an emulsion. In the second stage of the procedure, 200 µL samples of each plant extract were transferred to test tubes and a 5 mL aliquot of the aforementioned β-carotene emulsion was added to each. For the control assays, 200 µL of ethanol were used instead of the plant extract. Finally, all samples and controls were vortexed and submitted to 50 °C for 2 h in a heating block. The resultant absorbances were measured at 470 nm. For the plant extracts, the measurement was performed after the 2 h heating period, whilst in the case of the control assays it was made both immediately after the addition of the emulsion to the ethanol and after 2 h of incubation.

The antioxidant activity coefficient (AAC) was calculated using the following Equation (2):(2)$$\mathrm{Antioxidant}\;\mathrm{activity}\;\mathrm{coefficient}\;\left(\mathrm{AAC}\right)=\frac{As2-Ac2}{Ac0-Ac2}\times 100$$

*As*2 is the absorbance of the sample after the 2 h heating period, whilst *Ac*0 and *Ac*2 are the absorbances of the controls at time 0 and after 2 h of incubation, respectively.

### 3.6. Contaminants

#### 3.6.1. Mycotoxins

##### Extraction Procedure

The extraction of AFs, i.e., AFB1, AFB2, AFG1, AFG2, and OTA, from the samples was carried out according to the method described in our previous work [44]. Briefly, samples of approximately 2 g were weighed into centrifuge tubes and 100 µL of the internal standard ZAN was added from a solution at a concentration of 10 µg/mL. Thereafter, the samples were treated with 10 mL of acetonitrile (80%, *v*/*v*) and the tubes containing the mixture were placed on a shaker for 60 min at 110 rpm. Next, a centrifugation step was carried out at 12,669× *g* for 10 min, and after solid–liquid phase separation the supernatant layer was transferred to new tubes. Note that the extraction, centrifugation, and supernatant layer collection procedures were repeated, and all resulting supernatants were combined at the end of the process. Finally, an 8 mL aliquot of the obtained extract was evaporated, the residue redissolved in 1 mL acetonitrile (40%, *v*/*v*), and the solution filtered through 0.2 µm PVDF mini-uniprep filters prior to chromatographic analysis.

##### Ultra-High Performance Liquid Chromatography Coupled with Time-of-Flight Mass Spectrometry (UHPLC-ToF-MS) Analysis

An ultra-high performance liquid chromatograph (UHPLC) Nexera X2 (Shimadzu, Kyoto, Japan) coupled with a time-of-flight mass spectrometer (ToF-MS) AB Sciex triple TOF^TM^ 5600^+^ (Sciex, Foster City, CA, USA) was used for the separation and quantification of AFs and OTA. The aforementioned mycotoxins were run through a gradient on a Zorbax Eclipse Plus C18 column (2.1 × 50 mm, 1.8 µm particle size) from Agilent (Santa Clara, SA). The mobile phase system consisted of 0.1% formic acid in water (solvent A) and acetonitrile (solvent B). The optimised gradient elution procedure was as follows: 0–12 min, 10% solvent B; 12–13 min, 10–90% solvent B; 13–14 min, 90% solvent B; 14–15 min, 90–10% solvent B; 17–17 min, 10% solvent B. The injection volume was 20 µL, the flow rate was fixed at 0.5 mL/min, and the column temperature was maintained constant at 30 °C. The quantitative analysis was conducted by the peak area method, and the monitored ions were the protonated molecule [M + H]+ at m/z 313.07066 for AFB1, 315.08631 for AFB2, 329.06558 for AFG1, 331.08123 for AFG2, and 321.1696 for ZAN. The system was operated in the positive ion electrospray mode, and the MS parameters were set as follows: ion spray voltage: 5500 V; temperature: 575 °C; declustering potential: 100 V; curtain gas: 30 psig; nebuliser gas (gas 1) and heater gas (gas 2): 55 psig; mass range: 100–750 Da. Note that the method described was validated for linearity, limit of detection (LOD), limit of quantification (LOQ), accuracy and precision, and meets the requirements established by regulation (EC) no. 401/2006 [94]; for more details on the validation of the analytical method please refer to Lopes et al. [44]. The results obtained regarding the occurrence of AFB1, AFB2, AFG1, AFG2, and OTA were expressed in micrograms per kilogram of dried plant (aerial parts) (μg/Kg dp).

#### 3.6.2. Essential and Non-Essential Heavy Metals

Iron was determined according to the same methodology used for Ca, Mg, and P (see Section 3.4.2 Mineral Profile). Cu, Zn, Mn, Cr, Ni, and Co were determined by flame atomic absorption spectrometry (FAAS), while Cd and Pb were quantified by graphite furnace atomic absorption spectrometry (GFAAS). The equipment used for this analysis was a PinAAcle 900T spectrophotometer (Perkin Elmer Inc., Waltham, MA, USA). Hg levels were determined using an AMA254 mercury analyser (LECO instruments Ltd., St Joseph, MI, USA).

The LODs were Fe: 0.0512; Cu: 0.022; Zn: 0.005; Mn: 0.016; Cd: 0.010; Pb: 0.154; Cr: 0.022; Ni: 0.039; Co: 0.035; and Hg: 0.001 mg/L. The LOQs were Fe: 0.155; Cu: 0.068; Zn: 0.017; Mn: 0.048; Cd: 0.031; Pb: 0.469; Cr: 0.067; Ni: 0.117; Co: 0.107; and Hg: 0.005 mg/L. Note that for the determination of Cu, Zn, Mn, Cd, Pb, Cr, Ni, Co, and Hg the dry mineralisation method was used, in which approximately 1 g of each dehydrated and ground sample was incinerated in a muffle furnace and the obtained residues were dissolved in 5 mL of HCl (20%, *v*/*v*). Thereafter, the plant extract stock solutions were prepared by filtering into 50 mL volumetric flasks and adjusting the volume with deionised water. Finally, the target elements were measured in the obtained extract solutions. Calibration standard solutions were prepared from 1000 mg/L single element standard stock solutions by suitable dilution with deionised water. The obtained results were expressed in milligrams of element per kilogram of plant (aerial parts) on a dry weight basis (mg/Kg dw).

### 3.7. Statistical Analysis

The results were analysed by Student’s *t*-test to determine differences between *S. ramosissima* and *S. perennis alpini*, with *p* < 0.05 considered a significant difference. This statistical treatment was conducted with SPSS 26.0 statistical package for Windows (SPSS Inc., Chicago, IL, USA).

## 4. Conclusions

Both S. *ramosissima* and *S. perennis alpini* prove to be valuable sources of nutrients—in particular minerals and fibre—and bioactive compounds—i.e., phenolic acids and flavonoids—which makes these halophytes highly eligible for application in the agro-food and pharmaceutical industries. In particular, *S. ramosissima* is remarkable for its phenolic richness and antioxidant potential, and can be used directly as a functional food or ingredient, or as extract, or even as a source of isolated/individual phenolic compounds. These can be applied as natural additives and components of bioactive packaging, thus taking advantage of their antioxidant potential, hence providing innovative and healthier products for consumers while contributing to the reduction of food waste by extending the shelf life of other food products. Furthermore, the development of new cosmeceuticals, nutraceuticals, and pharmacological agents could be an interesting field of application for these halophytes. From another point of view, since both *Salicornia* and *Sarcocornia* can be cultivated in very hostile environments, by exploiting marginal resources such as saline soils and brackish water, their cultivation presents a key strategy to improve nutrition globally, while—particularly in more arid regions—simultaneously providing an improvement in the economic well-being of the population. Meanwhile, food safety, nutrition, and food security are strongly interconnected, and controlling the presence of contaminants, namely mycotoxins and HMs, is essential to ensure the protection of the health and safety of consumers.

One of the greatest dilemmas of this century by far is meeting food security needs while preserving global sustainability, a challenge in which the cultivation of halophytes can undoubtedly play a major role, but for all this potential to really bear fruit, it is essential that adequate cultivation strategies be implemented, preferably in controlled environments. Producers should be suitably trained, so that their proper knowledge of the particularities of halophytes can guarantee the high quality and safety of these, and awareness should be raised among consumers about the benefits of these species, in terms of health and sustainability.

Finally, more studies should be conducted on these species, with specimens collected at different times of the year, in different years, and from other areas of Portugal and other countries, to assess the influence of these factors on their composition.

## Figures and Tables

**Figure 1 molecules-28-02726-f001:**
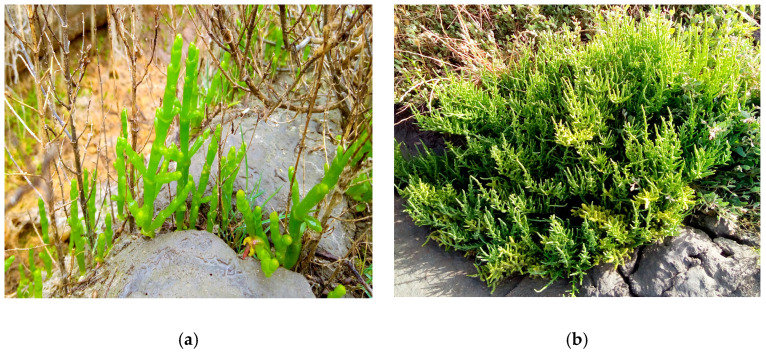
Overview of the halophytes studied. (**a**) *Salicornia ramosissima* (July); (**b**) *Sarcocormia perennis alpini* (July).

**Table 1 molecules-28-02726-t001:** Moisture (% fw) and nutrient composition (% dw) of *Salicornia ramosissima* and *Sarcocornia perennis alpini* (mean ± SD; *n* = 3).

Parameter	Content	
	*S. ramosissima*	*S. perennis alpini*
Moisture	89.7 ± 0.1 ^a^	87.8 ± 0.0 ^b^
Crude protein	6.61 ± 0.33 ^a^	4.28 ± 0.72 ^b^
Lipids	1.32 ± 0.14 ^a^	1.52 ± 0.08 ^a^
Crude fibre	11.3 ± 0.3 ^a^	15.3 ± 0.1 ^b^
Ash	39.5 ± 0.3 ^a^	40.4 ± 0.4 ^b^

In each row, different superscript letters denote significant differences (*p* < 0.05).

**Table 2 molecules-28-02726-t002:** Mineral composition (mg/g dw) of the aerial parts of *Salicornia ramosissima* and *Sarcocornia perennis alpini* (mean ± SD; *n* = 3).

Analytes	Content	
	*S. ramosissima*	*S. perennis alpini*
Sodium (Na)	204 ± 1 ^a^	177 ± 1 ^b^
Potassium (K)	6.71 ± 0.14 ^a^	8.08 ± 0.20 ^b^
Magnesium (Mg)	7.05 ± 0.12 ^a^	6.98 ± 0.17 ^b^
Phosphorous (P)	1.60 ± 0.00 ^a^	1.40 ± 0.09 ^b^
Calcium (Ca)	1.13 ± 0.23 ^a^	0.75 ± 0.00 ^a^

In each row, different superscript letters denote significant differences (*p* < 0.05).

**Table 3 molecules-28-02726-t003:** Total phenolic content (TPC) and total flavonoid content (TFC) of *Salicornia ramosissima* and *Sarcocornia perennis alpini* expressed in mg of gallic acid equivalents per g of extract dry weight (mg GAE/g extract dw) and mg of epicatechin equivalents per g of extract dry weight (mg ECE/g extract dw), respectively (mean ± SD; *n* = 3).

Parameter	Content	
	*S. ramosissima*	*S. perennis alpini*
Total phenolic content (TPC)	67.1 ± 1.7 ^a^	38.2 ± 1.5 ^b^
Total flavonoid content (TFC)	186 ± 3 ^a^	99.3 ± 0.8 ^b^

In each row, different superscript letters denote significant differences (*p* < 0.05).

**Table 4 molecules-28-02726-t004:** Identification and tentative identification of the phenolic compounds in the extracts of *Salicornia ramosissima* and *Sarcocornia perennis alpini* by UHPLC-PDA-ESI-MS/MS.

#	t_r_ (min)	λ_max_ (nm)	[M]^-^ *m*/*z*	Main Products *m*/*z*	Molecular Formula	Structural Subclass	Tentative Identification	*S. ramosissima*	*S. perennis alpini*	Confirmation/Ref. ^1^
1	3.14	325	343	191	C_14_H_16_O_10_	Hydroxybenzoic acids	3-Galloylquinic acid	✓	-	[34,35]
2	4.0	293	153	109, 108	C_7_H_14_O_7_	Hydroxybenzoic acids	Protocatechuic acid	✓	✓	[11,35], Std ^2^
3	5.67	325	353	191, 179	C_16_H_18_O_9_	Hydroxycinnamic acids	Neochlorogenic acid	✓	✓	[11,34,35]
4	8.94	325	179	134, 135	C_9_H_8_O_4_	Hydroxycinnamic acids	Caffeic acid	✓	✓	[11,13,14,36], Std ^2^
5	9.65	325	353	191.2, 179.0, 93.20	C_16_H_18_O_9_	Hydroxycinnamic acids	Chlorogenic acid	✓	✓	[11,13,14], Std ^2^
6	9.93	330	355	193	C_16_H_20_O_9_	Hydroxycinnamic acids glycosides	Ferulic acid-*O*-hexoside	✓	✓	[35,37]
7	10.04	325	337	173	C_16_H_18_O_8_	Hydroxycinnamic acids	5-Coumaroylquinic acid derivate	✓	✓	[34,35]
8	11.66	330	385	223	C_17_H_22_O_10_	Hydroxycinnamic acids glycosides	Sinapic acid-*O*-hexoside	✓	✓	[37]
9	12.43	325	337	173	C_16_H_18_O_8_	Hydroxycinnamic acids	4-Coumaroylquinic acid derivate	✓	-	[34,35]
10	12.51	310	163	119, 93	C_9_H_8_O_3_	Hydroxycinnamic acids	*p*-Coumaric acid	✓	✓	[11,13,14,35,36], Std ^2^
11	12.70	325	343	191	C_14_H_16_O_10_	Hydroxybenzoic acids	3-Galloylquinic acid	-	✓	[34,35]
12	14.61	325	193	134	C_10_H_10_O_4_	Hydroxycinnamic acids	Ferulic acid	✓	✓	[11,13,14,36], Std ^2^
13	20.87	356	771	300	C_33_H_40_O_21_	Flavonol glycosides	Quercetin-3-*O*-glucosylrutinoside	✓	-	[34]
14	21.23	356	463	300, 271	C_21_H_20_O_12_	Flavonol glycosides	Isoquercitrin	✓	✓	[13,34]
15	21.30	356	609	300, 271	C_27_H_30_O_16_	Flavonol glycosides	Rutin	✓	✓	[11,13,35,38], Std ^2^
16	21.70	350	433	301, 271	C_21_H_22_O_10_	Flavonol glycosides	Naringenin-7-*O*-glucoside	✓	✓	[11]
17	21.92	365	317	151, 179	C_15_H_10_O_8_	Flavonols	Myricetin	✓	✓	[13,14,35], Std ^2^
18	22.29	356	447	285	C_21_H_20_O_11_	Flavonol glycosides	Luteolin-7-*O*-glucoside	✓	✓	[11]
19	22.30	528	579	271	C_27_H_32_O_14_	Flavanone glycosides	Narirutin	✓	-	[35]
20	22.31	356	593	285	C_27_H_30_O_15_	Flavonol glycosides	Kaempferol-3-*O*-rutinoside	✓	✓	[34]
21	22.48	350	477	301, 300	C_21_H_20_O_11_	Flavonol glycosides	Quercitrin	✓	✓	[37]
22	22.87	356	447	285	C_21_H_20_O_11_	Flavonol glycosides	Kaempferol-3-*O*-glucoside	✓	-	[34]
23	22.90	350	431	269, 151	C_21_H_20_O_10_	Flavone glycosides	Apigenin-7-glucoside	✓	✓	[13,38]
24	23.13	295	271	151, 119	C_15_H_12_O_5_	Flavanones	Naringenin	✓	✓	[13], Std ^2^
25	23.34	360	-	88, 70	C_15_H_10_O_7_	Flavonols	Quercetin	✓	✓	[11,13,14,36], Std ^2^
26	23.81	350	285	151, 133, 132	C_15_H_10_O_6_	Flavones	Luteolin	✓	✓	[13,38], Std ^2^
27	24.49	367	285	185, 151, 93	C_15_H_10_O_6_	Flavonols	Kaempferol	✓	✓	[13,14,38], Std ^2^
28	24.72	360	269	151, 149, 117	C_15_H_10_O_5_	Flavones	Apigenin	✓	✓	[13,36], Std ^2^
29	25.21	278	577	289	C_30_H_26_O_12_	Flavanols	Procyanidin B-type	✓	✓	[35]
30	26.60	278	577	407, 289	C_30_H_26_O_12_	Flavanols	Procyanidin B-type	✓	-	[35]
31	30.79	370	315	151	C_16_H_12_O_7_	Flavonols	Isorhamnetin	-	✓	[14]
32	33.80	350	431	269, 151	C_21_H_20_O_10_	Flavone glycosides	Apigenin-7-glucoside	✓	✓	[13,38]
33	35.09	528	579	271	C_27_H_32_O_14_	Flavanone glycosides	Naringin	✓	✓	[35]

t_r_—retention time; λ_max_—maximum absorption wavelengths; [M]^−^—molecular ions; ✓—indicates the presence of the compound identified; ^1^ Ref.—references used to support tentative identification of compounds or ^2^ Std (standards available) to confirm the identification.

**Table 5 molecules-28-02726-t005:** Detected parameters: SRM, linearity, LOD, and LOQ; and individual and total amounts of the main phenolic compounds (µg/g extract) quantified in *Salicornia ramosissima* and *Sarcocornia perennis alpini*.

Peak #	Phenolic Compound	SRM Transition	Equation	Conc. Range (µg/mL)	r^2^	LOD (µg/g Extract fw)	LOQ (µ/g Extract fw)	Content			
								*S. ramosissima* (µg/g Extract)		*S. perennis alpini* (µg/g Extract)	
								fw	dw	fw	dw
**Phenolic acids**											
2	Protocatechuic acid	153 → 109	y = 361,934.7x − 2718.23	0.01 − 1	0.9996	0.09	0.19	13.68	133.1	2.882	23.68
4	Caffeic acid	179 → 135	y = 1,291,832x − 3938.19	0.01 − 1	0.9994	0.09	0.19	17.02	165.6	0.956	7.855
5	Chlorogenic acid	353 → 191	y = 1,063,638x + 34221.2	0.01 − 1	0.9964	0.08	0.19	23.04	224.1	12.24	100.6
10	*p*-Coumaric acid	163 → 119	y = 114,972.9x − 5424.57	0.05 − 1	0.9981	0.48	0.97	175.0	1702	55.57	456.6
12	Ferulic acid	193 → 134	y = 13,102.51x − 3267.00	0.05 − 1	0.9927	0.48	0.97	<LOQ	<LOQ	<LOQ	<LOQ
Σ Phenolic acids ^1^								228.8	2225	71.65	588.7
**Flavonoids**											
15	Rutin	609 → 300	y = 2,259,218x − 2927.46	0.001 − 1	0.9999	0.01	0.02	41.94	408.0	19.47	160.0
17	Myricetin	317 → 151	y = 370,328x − 27222.2	0.05 − 1	0.9957	0.48	0.97	17.16	166.9	14.71	120.9
24	Naringenin	271 → 151	y = 1,667,721x − 1606.87	0.001 − 1	0.9930	0.01	0.02	0.573	5.574	0.085	0.698
25	Quercetin	274 → 88	y = 60,716.41x − 1363.65	0.001 − 1	0.9996	0.01	0.02	93.00	904.7	68.52	563.0
26	Luteolin	285 → 133	y = 2,663,392x − 80.951	0.001 − 1	0.9997	0.01	0.02	1.698	16.52	0.611	5.021
27	Kaempferol	285 → 151	y = 254,318.2x − 2752.38	0.0025 − 1	0.9975	0.025	0.05	23.44	228.0	6.303	51.79
28	Apigenin	269 → 117	y = 1,006,111x − 778.086	0.001 − 1	0.9921	0.01	0.02	<LOQ	<LOQ	<LOQ	<LOQ
Σ Flavonoids ^1^								177.8	1730	109.7	901.4
Σ Total ^2^								406.6	3955	181.3	1490

SRM—selected reaction monitoring; r^2^—coefficient of determination; LOD—limit of determination; LOQ—limit of quantification; dw—dry weight; fw—fresh weight; ^1^ The sum (Σ) of each class of phenolic compounds and ***^2^*** the total content are highlighted in bold.

**Table 6 molecules-28-02726-t006:** Antioxidant capacity of *Salicornia ramosissima* and *Sarcocornia perennis alpini*. DPPH radical inhibition assay expressed in mg of trolox equivalents per g of extract dry weight (mg TE/g extract dw) and β-carotene bleaching inhibition test expressed as the antioxidant activity coefficient (AAC) (mean ± SD; *n* = 3).

Parameter	Content	
	*S. ramosissima*	*S. perennis alpini*
DPPH	30.2 ± 1 ^a^	11.0 ± 0.4 ^b^
β-carotene bleaching inhibition	1697 ± 82 ^a^	1403 ± 57 ^b^

In each row, different superscript letters denote significant differences (*p* < 0.05).

**Table 7 molecules-28-02726-t007:** Mycotoxin contamination (μg/Kg dp) in *Salicornia ramosissima* and *Sarcocornia perennis alpini* (mean ± SD; *n* = 2).

Parameter	Content	
	*S. ramosissima*	*S. perennis alpini*
AFB1	5.21 ± 0.06	n.d.
AFB2	n.d.	n.d.
AFG1	n.d.	n.d.
AFG2	n.d.	n.d.
Total AFs	5.21	n.d.
OTA	n.d.	n.d.

n.d.—not detected. LOD: 0.5 µg/Kg for AFB1, AFB2, and AFG1; 2 µg/Kg for AFG2; and 3 µg/Kg for OTA [44]. LOQ: 1 µg/Kg for AFB1, AFB2, and AFG1; 3 µg/Kg for AFG2; and 4.5 µg/Kg for OTA [44].

**Table 8 molecules-28-02726-t008:** Heavy metals contamination (mg/Kg fw) in *Salicornia ramosissima* and *Sarcocornia perennis alpini* (mean ± SD; *n* = 3).

Analytes	Content	
	*S. ramosissima*	*S. perennis alpini*
Copper (Cu)	1.11 ± 0.17 ^a^	1.31 ± 0.05 ^a^
Zinc (Zn)	2.01 ± 0.04 ^a^	2.61 ± 0.03 ^b^
Manganese (Mn)	6.54 ± 0.15 ^a^	4.63 ± 0.05 ^b^
Iron (Fe)	n.d.^1^	n.d.^1^
Cadmium (Cd)	0.02 ± 0.00 ^a^	0.01 ± 0.00 ^a^
Lead (Pb)	0.01 ± 0.00 ^a^	0.02 ± 0.00 ^a^
Chromium (Cr)	1.32 ± 0.17 ^a^	1.45 ± 0.14 ^a^
Nickel (Ni)	0.87 ± 0.05 ^a^	1.75 ± 0.14 ^b^
Cobalt (Co)	1.03 ± 0.03 ^a^	0.92 ± 0.01 ^b^
Mercury (Hg)	n.d.^2^	0.01 ± 0.00

n.d.—Not detected. ^1^ LOD: 0.0512 mg/L; ^2^ LOD: 0.001 mg/L. In each row, different superscript letters denote significant differences (*p* < 0.05).

## Data Availability

Data are contained within the article.

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
