# Peer review of "Towards the Sustainable Exploitation of Salt-Tolerant Plants: Nutritional Characterisation, Phenolics Composition, and Potential Contaminants Analysis of Salicornia ramosissima and Sarcocornia perennis alpini"

_molecules, 2023, doi:10.3390/molecules28062726_

Round 1

Reviewer 1 Report

This work provides a detailed analysis of the nutritional comosition and potential contaminants of Salicornia ramosissima and Sarcocornia perennis alpine, which is of interest for the exploitation of these two halophytes. The manuscript was well organized, and the results were presented logically and discussed fully. Some minor points to improve the manuscript.

1)      Table 5: Is there any replicate for each analysis? Please add the results of statistical analysis. Additionally, the parameters of detection SRM, linearity, LOD and LOQ can be deleted.

2)      The data of inhibition percentage for DPPH assay is unnecessary.

Reviewer 2 Report

The Manuscript “Towards the sustainable exploitation of salt-tolerant plants: Nutritional characterisation, phenolics composition, and potential contaminants analysis of Salicornia ramosissima and Sarcocornia perennis alpini“ is an interesting article presenting an examination of plants’ nutritional, phytochemical potential as well as contamination level. The methods are scientifically sound and the results are relevant.

However, the manuscript is written lengthily, with a lot of literature data cited for comparison, making it difficult to read. I suggest that authors present literature data in table(s) and make the presentation clearer and more interesting. Also, the Materials and Methods section is too extensive. Methods should be described briefly, avoiding too many explanations. Instead, I suggest citing the literature for more detailed instructions.

Generally, in my opinion, the manuscript needs major revision before taking into consideration for publication.
